# Robotic Complete ALPPS (rALPPS)—First German Experiences

**DOI:** 10.3390/cancers16051070

**Published:** 2024-03-06

**Authors:** Jörg Arend, Mareike Franz, Alexander Rose, Christine March, Mirhasan Rahimli, Aristotelis Perrakis, Eric Lorenz, Roland Croner

**Affiliations:** 1Department of General-, Visceral-, Vascular- and Transplant Surgery, University Hospital Magdeburg, 39120 Magdeburg, Germany; mareike.franz@med.ovgu.de (M.F.); alexander.rose@med.ovgu.de (A.R.); mirhasan.rahimli@med.ovgu.de (M.R.); aristotelis.perrakis@med.ovgu.de (A.P.); eric.lorenz@med.ovgu.de (E.L.); roland.croner@med.ovgu.de (R.C.); 2Department of Radiology and Nuclear Medicine, University Hospital Magdeburg, 39120 Magdeburg, Germany; christine.march@med.ovgu.de

**Keywords:** robotic, hepatectomy, liver surgery, ALPPS, minimally invasive liver surgery

## Abstract

**Simple Summary:**

Minimally invasive surgery is a decisive step forward in reducing morbidity and mortality in surgery. In liver surgery, this procedure is already standard for small resections. Complex operations on the liver are only performed in a few experienced centers. ALPPS is a very complex procedure and requires the highest level of expertise. Conventional ALPPS is associated with very high morbidity and mortality. Our clinic has great expertise in robotic liver surgery, particularly in the area of major liver resections. For this reason, we have started with the complete robotic ALPPS and would like to present our first experiences. From our point of view, the robotic approach is particularly advantageous for this procedure. As a result, morbidity and mortality can be minimized with faster convalescence.

**Abstract:**

Background: ALPPS leads to fast and effective liver hypertrophy. This enables the resection of extended tumors. Conventional ALPPS is associated with high morbidity and mortality. MILS reduces morbidity and the robot adds technical features that make complex procedures safe. Material and Methods: The MD-MILS was screened for patients who underwent rALPPS. Demographic and perioperative data were evaluated retrospectively. Ninety days postoperative morbidity was scored according to the CD classification. The findings were compared with the literature. Results: Since November 2021, five patients have been identified. The mean age and BMI of the patients were 50.0 years and 22.7 kg/m^2^. In four cases, patients suffered from colorectal liver metastases and, in one case, intrahepatic cholangiocarcinoma. Prior to the first operation, the mean liver volume of the residual left liver was 380.9 mL with a FLR-BWR of 0.677%. Prior to the second operation, the mean volume of the residual liver was 529.8 mL with a FLR-BWR of 0.947%. This was an increase of 41.9% of the residual liver volume. The first and second operations were carried out within 17.8 days. The mean time of the first and second operations was 341.2 min and 440.6 min. The mean hospital stay was 27.2 days. Histopathology showed the largest tumor size of 39 mm in diameter with a mean amount of 4.7 tumors. The mean tumor-free margin was 12.3 mm. One complication CD > 3a occurred. No patient died during the 90-day follow up. Conclusion: In the first German series, we demonstrated that rALPPS can be carried out safely with reduced morbidity and mortality in selected patients.

## 1. Background

For primary and secondary liver tumors, resection is the curative treatment of choice in an interdisciplinary setting [1,2,3,4,5,6]. In this course, minimally invasive liver surgery (MILS) should be the selected procedure. MILS results in all perioperative benefits as described during other minimally invasive procedures compared to conventional surgery. This includes lower postoperative morbidity, shorter hospitalization, lower rates of postoperative liver functional impairment, as well as rapid mobilization and transition to a normal diet by maintaining equal oncological results compared to open surgery [7,8,9,10,11]. Only complete tumor resection (R0) offers the chance of long-term survival and cure for the patient. But this can be achieved only if a future liver remnant (FLR) of 25–30% and an impaired parenchyma of 40% can be preserved [12]. The future liver volume to body weight ratio (FLV-BWR) can be calculated for a better assessment of the perioperative risk. A FLV-BWR of >0.5% results in a significant improvement in postoperative morbidity and mortality [13]. For bigger lesions with bilobular spread and expected small remnant liver volume, primary resection is usually challenging. Several strategies are described to increase the rate of resectability and to prevent postoperative liver failure. Interventional embolization of the portal vein leads to 20–46% hypertrophy in the course of 6–8 weeks, but there is a remaining risk of tumor progression without tumor therapy at the time of liver hypertrophy [4,14]. In contrast to portal vein embolization, the Associating Liver Partition and Portal Vein Ligation (ALPPS) enables a timely and sufficient enlargement of the future remnant liver volume (FLV) [15]. ALPPS as an open procedure is associated with high morbidity and mortality. To reduce the operative trauma and associated morbidity and mortality, MILS could be a strategy for ALPPS. Complete robotic Associating Liver Partition and Portal Vein Ligation (rALPPS) is performed in a few very specialized centers only. The current robotic systems offer technical advantages compared to conventional laparoscopy such as an EndoWrist^®^ with full mobility, stable 3D visualization, and the possibility of precise dissection under vessel control [16,17,18,19,20]. Hereby, the morbidity and mortality for rALPPS are in the range of 13.6–64% and 0–29%, respectively [4,15,21]. Because of increasing multimodal preoperative treatment strategies, the rate of parenchymal damage of the liver increases. Therefore, besides FLV and the calculated FLV-BWR, monitoring of liver function is inevitable. The LiMAx^®^ is one established method for liver-specific functional assessment [22]. Only the combination of remnant liver volume and liver function enables a realistic assessment of the perioperative risk for liver failure. Another requirement for rALPPS is the intraoperative visualization of the tumor to define the resection margins. Therefore, intraoperative ultrasound is inevitable and can be combined with fluorescence tumor visualization with Indocyanine green (ICG) to enhance precision [23]. Hereby, a complete and safe oncological resection (R0) can be achieved.

## 2. Material and Methods

### 2.1. Patients

The Magdeburg registry of minimally invasive liver surgery (MD-MILS) was screened for patients who underwent robotic associated liver partition and portal vein ligation (rALPPS). Patient and tumor characteristics, and peri- and intraoperative data of these patients were collected prospectively and analyzed retrospectively. Postoperative morbidity was described using the Clavien–Dindo classification and the 90-day postoperative mortality was evaluated. Since this is a retrospective study, IRB approval is not required.

### 2.2. Selection Criteria for rALPPS

Robotic ALPPS was considered for patients who suffered from primary or secondary liver malignancies without any vessel infiltration. The tumor location must have indicated an extended hemi-hepatectomy without tumor seedings in the remnant liver segments in preoperative diagnostics, which includes liver MRI for all cases. Patients who needed multi-visceral resection or vessel reconstruction were not selected for rALPPS. Liver function was assessed preoperatively with a LiMAX-Test to exclude patients with marginal hepatic synthesis who were at risk of inappropriate liver hypertrophy. The future liver volume (FLV) was calculated using the future liver volume–body weight ratio.

### 2.3. Technique of rALPPS

Patients were placed in a reversed Trendelenburg position. The robotic trocars were placed in the middle of the abdomen, approx. 15 cm below the costal arch, as described elsewhere [24,25]. The trocars were inserted in the right upper abdomen, to the right and left of the navel, and in the left upper abdomen in a single line. One laparoscopic trocar for table assistance and pringle maneuver was placed between robotic trocars 2 and 3 and in the left lateral abdomen. Then, intraoperative ultrasound was performed to identify the liver tumor burden. Resection margins on the liver surface were marked with ultrasound assistance under near-infrared fluorescence using indocyanine green (ICG), which was applied prior to surgery [23]. The cranial proportion of the inferior vena cava (IVC) was identified. After that, the hepato-duodenal ligament was surrounded with a pringle maneuver using a thoracic tube and umbilical tape as described previously [24]. Then, the dissection of the hepato-duodenal ligament was initiated. In the case of primary liver tumors (ICC), a lymph node dissection of the ligament that was extended along the common hepatic artery to the celiac trunk was performed. The right hepatic artery was identified and surrounded by a vessel loop, which was clipped at the end and left in place. After this, the portal vein and the bifurcation in the left and right hepatic branches were visualized. The right portal vein branch was clipped and divided. Then, the parenchyma dissection between liver segments II/III and IV was performed using the three-device technique until the ventral border of the IVC [16]. Intrahepatic bile ducts and vessels were clipped and divided selectively. The right hepatic bile duct was left in place. Between the divided liver fragments, TachoSil^®^ (Takeda, Berlin, Germany) was placed to prevent adhesions during the second staged operation, and an abdominal drain was placed.

The second operation to complete rALPPS followed the first step with a mean time of 16 days after appropriate hypertrophy of the remnant liver, which was controlled by MRI, calculating the FLV-BWR. The trocars were placed using the initial positions at the abdomen from the first operation. The teres and falciform ligaments were dissected from the ventral wall of the abdomen. The looped right hepatic artery was identified, clipped, and divided. Then, the liver was mobilized from the IVC. Hereby, the Spigelian veins were clipped and dissected, and the Makuuchi ligament was divided. Then, the parenchyma dissection plane was reopened, and the right hepatic bile duct was clipped and divided. The right hepatic vein was identified and cut with a vascular stapler. The specimen was placed in a retrieval bag and removed via a Pfannenstiel incision. Finally, the perfusion of the remnant liver was controlled with an intraoperative ultrasound and an abdominal drain was placed (Figure 1).

### 2.4. Statistical Analysis

Data were analyzed with IBM SPSS Statistics für Windows, version 28 (IBM Corp., Armonk, NY, USA). The data are presented as mean ± standard deviation (SD) (Table 1 and Table 2). Pubmed research was performed including reports regarding rALPPS until December 2022. The data from the literature were compared with data from the MD-MILS registry (Table 3 and Table 4).

## 3. Results

### 3.1. Patient Characteristics

Four female patients and one male patient with a mean age of 50.0 ± 8.5 years and a mean body mass index of 22.7 ± 3.6 who underwent rALPPS were identified in the MD-MILS. The ASA classification was two in all cases and the mean LiMAx value prior to liver surgery was 485.3 ± 83.5 µg/kg/h. Four patients suffered from colorectal liver metastases (CRLM), and one patient suffered from intrahepatic cholangiocarcinoma (ICC). The mean removed liver mass was 780.2 g, the mean tumor size was 45.4 mm, the mean tumor burden was 4.2 tumors on the affected liver side, and the smallest tumor-free resection margin was 9.5 mm. In four cases, neoadjuvant chemotherapy was performed prior to rALPPS (Table 1).

### 3.2. Perioperative Parameters

Hepatic enzymes were measured within the first postoperative days. The peak of the GLDH was at postoperative day three (1523.2 nmol/sl) after the first step and on postoperative day one (4741.5 nmol/sl) after the second step of rALPPS. ALAT and ASAT peaked on the first postoperative day after the first and second operation (Figure 2).

The FLV through imaging was calculated with a mean of 380.9 ± 138.1 mL prior to the first and 529.8 ± 162.9 mL before the second operation. The FLV-BWR was 0.677 ± 0.196% before the first and 0.947 ± 0.223% before the second operation. This means that there was an enlargement of 41.9 ± 13.2% future liver volume and an increase of 42.0 ± 13.1% of the FLV-BWR after the first operation within a time period of 17.8 ± 5.4 days where the second step of rALPPS was performed (Figure 3).

The mean operation time of the first step was 341.0 ± 145.2 min and of the second step of rALPPS was 440.6 ± 158.4 min. The mean blood loss during the first part of the procedure was 140.0 ± 54.8 mL and during the second part was 660.0 ± 313.1 mL, with only one packed red blood cell transfusion in one case (Table 2). In all cases, specimen margins were tumor free during histopathological examination, with a mean margin of 9.5mm (Table 1).

The mean overall postoperative hospital stay was 27.2 ± 7.7 days. There was one major complication of Clavien–Dindo > 3a (1/10 operations, 10%) within 90 days. This was a bile leak that could be handled by reintervention. No patient died during the 90 days of follow up (Table 2).

### 3.3. Comparison with the Literature

Screening the literature, 8 reports, including 10 patients, could be identified regarding rALPPS. In six cases, the patients suffered from hepatocellular carcinoma, in two cases, from colorectal liver metastases, and, in one case, from intrahepatic cholangiocarcinoma. Regarding Europe, only reports from Spain and Italy were found for rALPPS (Table 3).

The operation time of the first step ranged from 195.0 to 540.0 min (mean 407.0 min) and 180.0 to 380.0 min (mean 251.0 min) in the second step of rALPPS. The intraoperative blood loss in the first operation was 150.0–500.0 mL (mean 360.0 mL) and 200.0–500.0 mL (mean 328.0 mL) in the second operation of the procedure. An increase of 30–112% (mean 67%) of the future liver volume between steps one and two was described in the literature, which could be reached in a mean time period of 10–21 days (mean 15 days) between the operations. The mean postoperative hospital stay was 8–64 days (mean 21 days), as described in the evaluated reports (Table 4).

## 4. Discussion

Our data and the current data from the literature demonstrate that rALPPS is technically feasible and safe [26,27,28,29,30,31,32,33]. The oncological adequate resection of advanced liver tumors requires a high level of expertise. Therefore, complex procedures are performed minimally invasively in specialized centers only. For this reason, only case reports or mini-series for rALPPS currently exist. Conventional ALPPS is associated with high morbidity and mortality when proper risk assessment is not performed. Thus, numerous modifications for ALPPS such as mini ALPPS and hybrid ALPPS, and radiological procedures such as microwave or radiofrequency-assisted ALPPS have been described [34,35,36,37,38,39]. The aim of these modifications is to reduce surgical trauma and thus decrease perioperative morbidity and mortality [40]. Minimally invasive ALPPS by laparoscopy represents a decisive further development. It leads to a significant reduction in surgical trauma, abdominal adhesions between the surgical steps, and less intraoperative blood loss compared to the open technique [41]. The postoperative complication rate, and especially the liver failure rate, is decreased, which has a significant influence on postoperative mortality [42,43]. The feasibility and safety of laparoscopic ALPPS have been demonstrated in various studies [41,42,44,45]. Robotic use in liver surgery with expanded indications is increasing with proven safety and feasibility [46]. Regarding technical innovations, robotics has some advantages compared to conventional laparoscopy. This includes the EndoWrist, tremor elimination, ergonomics, 3D visualization, fluorescence tumor visualization, and the resulting precise dissection with high oncological quality [16,23]. In a meta-analysis published by Zhang et al., robotic versus laparoscopic liver surgery showed a longer operating time, but this was highly related to the learning curve. In addition, the average tumor diameter and the number of lesions were larger in the robotic vs. the laparoscopic group, which indicates more complex procedures during robotic surgery [47]. A recent analysis performed by Chong et al. elucidated a lower conversion rate (*p* = 0.01) and shorter length of stay (*p* = 0.04) for robotic vs. laparoscopic right hemi- or extended hemi-hepatectomy [48]. The difficulty score of the robotic procedures was significantly higher in this study [49]. In addition, Kamarajah et al. figured out a significantly lower readmission rate for robotic liver resections (*p* = 0.005) in their meta-analysis [50]. This analysis confirmed significantly less blood loss and a lower conversion rate for robotic vs. laparoscopic liver resections, although a higher rate of major procedures was performed in the robotic group. In a further meta-analysis including 485 patients, lower blood loss and conversion rates were confirmed for major liver resections for robotic compared to laparoscopic procedures [51]. The decrease in perioperative morbidity in robotics compared to laparoscopy reduces the hospital stay of patients and hereby even costs [52].

We present the first series of rALPPS from Germany. The rapid reconvalescence after the first procedure with timely and sufficient hypertrophy of the future liver remnant demonstrates feasibility, with low morbidity and lack of mortality. Furthermore, the results show that the 1st and 2nd steps of rALPPS could be performed with a mean blood loss of 140 mL and 660 mL, respectively, in a time of 341.2 and 440.6 min. These findings are in concordance with the current literature [26,27,28,29,30,31,32,33]. It has to be mentioned that we used the ERBEJET 2^®^ (Erbe, Tübingen, Germany) within the three-device technique for parenchymal dissection. The dilution with the irrigation fluid means that the blood loss determined might be even lower than that measured [16]. Blood transfusion was required intraoperatively in only one case (packed red blood cells). The mean time period between operations one and two was 17.8 days in our series. In the literature, an interval between 7.4 and 38.0 days is described [41,42]. The time frame between the two steps of ALPPS for minimally invasive procedures is almost twice as long as for the conventional technique (10 vs. 20 days). The prolonged waiting time allows for not only hypertrophy but also hyperplasia, and thus an improvement in function and a possible reduction in the rate of postoperative liver failure [42]. Overall, the data show that sufficient hypertrophy occurs promptly. With an increase in FLR of 41.9% and FLR-BWR of 42.0%, rALPPS is an effective procedure for inducing hypertrophy. In a systematic review from Baili et al., an increase in FLR of 68% (10.5–110%) at days 4–15 after the first step of ALPPS was identified [53]. They described a 30-day mortality of 9.55%. In our series, there was no 90-day mortality, with a perioperative morbidity of 10.0% (CD ≥ 3a). The morbidity reported in the literature is 13.6–64% [4,15,21]. Despite the complexity of the procedure, we can show a low rate of severe complications (CD ≥ 3a). Tumor-directed pre-treatment was performed in 4 cases (80.0%), and major surgery before rALPPS in three cases (60.0%). Previous abdominal surgery does not represent an increased preoperative risk for rALPPS [54]. In the present series of rALPPS, four cases were CRLM, and one case was intrahepatic CC. The mean number of tumor foci was 4.2 with a largest tumor diameter of 45.4 mm. Histological work-up revealed a smallest safety margin of 9.5 mm. Thus, rALPPS leads to a sufficient oncological quality even in advanced tumors. The oncological safety of minimally invasive liver surgery leads to comparable disease-free survival and overall survival compared to conventional liver surgery [55,56,57]. In a propensity score match analysis of Lim et al. with 111 laparoscopic and 61 robotic liver procedures, there was also no difference in 1-, 2- and 3-year DFS and OS [57].

## 5. Conclusions

Our experience identified rALPPS as a safe and sufficient procedure for advanced liver tumors. Due to having a two-step procedure, the first intervention, which can be performed robotically, is as gentle as possible. In the second step, a safe oncological resection can be performed due to the low postoperative adhesions and the precise dissection. Our initial data show that low morbidity without mortality can be achieved in this complex procedure thanks to a high level of expertise in robotic liver surgery.

## Figures and Tables

**Figure 1 cancers-16-01070-f001:**
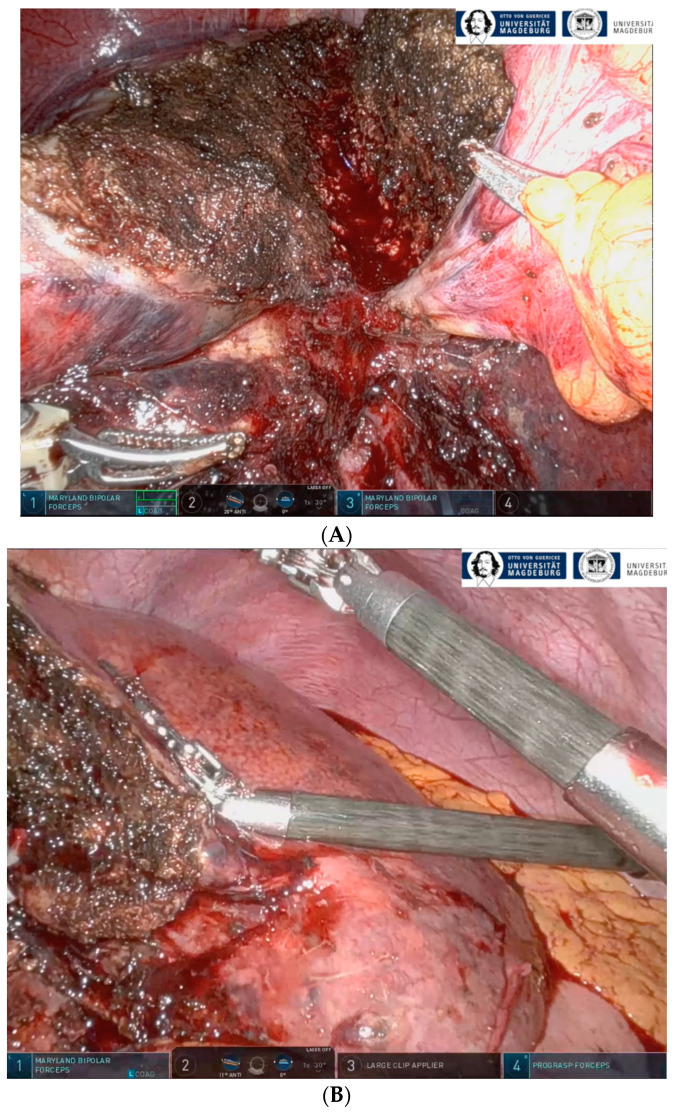
(**A**) The divided liver between liver segments II/III and IV after the first step of full robotic liver partition and portal vein ligation (rALPPS). (**B**) The remaining liver segments II/III after hypertrophy and completion of rALPPS after the second operation.

**Figure 2 cancers-16-01070-f002:**
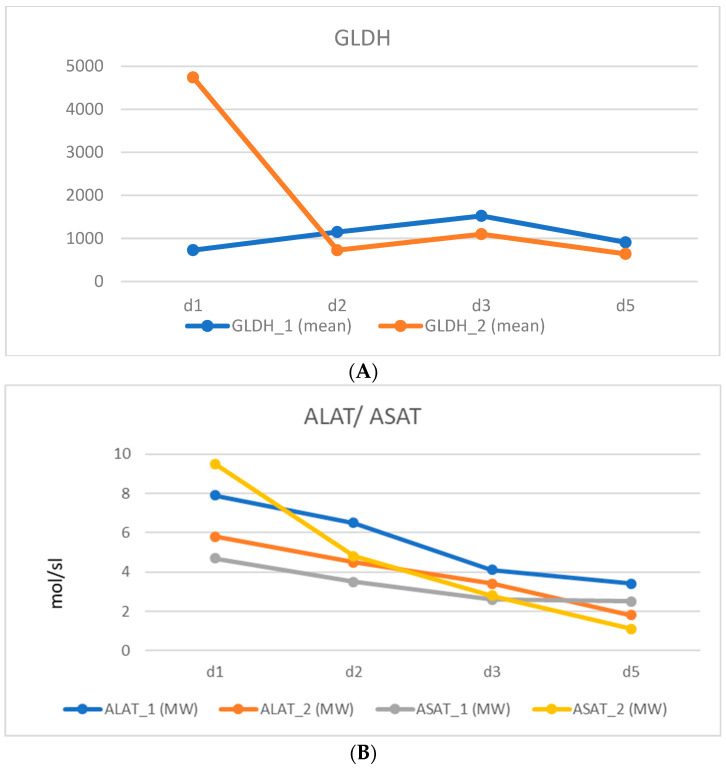
Postoperative liver enzymes (all cases, mean, d1-d5 postoperativ) (**A**) GLDH (nmol/sl); (**B**) ALAT/ASAT (µmol/sl) after the first (1) and second steps (2) of complete robotic associated liver partition and portal vein ligation (rALPPS).

**Figure 3 cancers-16-01070-f003:**
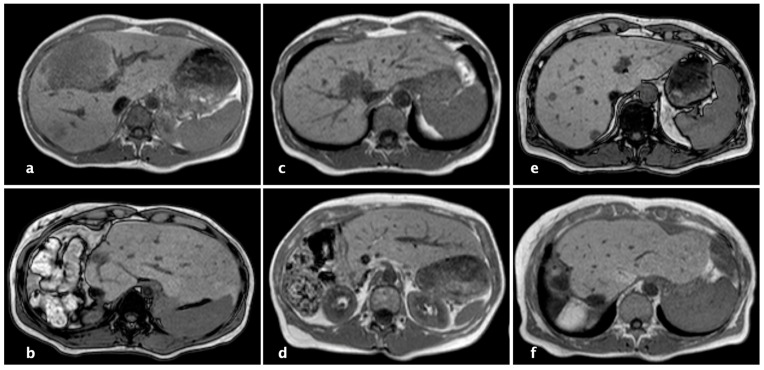
Magnet resonance imaging (MRI) of patients who underwent complete robotic-associated liver partition and portal vein ligation (rALPPS), (**a**,**c**,**e**): before rALPPS, (**b**,**d**,**f**): hypertrophy of the liver remnant after complete rALPPS.

**Table 1 cancers-16-01070-t001:** Patient demographics and tumor characteristics of patients who underwent robotic associated liver partition and portal vein ligation (rALPPS). CRLM: colorectal liver metastases, ICC: intrahepatic Cholangiocarcinoma, CTX: chemotherapy, SD: standard deviation.

**Total, n**	5
**Age; years, mean (SD)**	50.0 (8.%)
**Gender; female: n (%)**	4 (80)
**BMI; kg/m² (SD)**	22.7 (3.6)
**ASA classification (%)**	2 (100)
**LiMAx; µg/kg/h; mean (SD)**	485.3 (83.5)
**Neoadjuvant CTX; n (%)**	4 (80)
**Prior abdominal surgery; n (%)**	3 (60.0)
**Tumor**	
**a.** **CRLM, n (%)**	4 (80.0)
**b.** **ICC, n (%)**	1 (20.0)
**c.** **Size largest tumour; mm, mean (SD)**	45.4 (27.9)
**d.** **Amount of tumors; n (SD)**	4.2 (1.9)
**e.** **Tumor-free resection margin; mm (SD)**	9.5 (17.0)
**f.** **Removed liver mass; g (SD)**	780.2 (180.2)

**Table 2 cancers-16-01070-t002:** Perioperative outcomes of patients who underwent complete robotic-associated liver partition and portal vein ligation (rALPPS). FLV: future liver volume, BWR: body weight ratio, SD: standard deviation.

**FLV; mean, mL (SD)**	
** 1. Operation**	380.9 (138.1)
** 2. Operation**	529.8 (163.0)
**Hypertrophy; mean %, (SD)**	41.9 (13.2)
**FLV-BWR**	
** 1. Operation**	0.677 (0.196)
** 2. Operation**	0.947 (0.223)
**FLV-BWR Increase; mean %, (SD)**	42.0 (13.1)
**Time from 1. to 2. operation; d (SD)**	17.8 (5.4)
**OR time; min (SD)**	
** 1. Operation**	341.2 (145.2)
** 2. Operation**	440.6 (158.4)
**Blood loss; mL (SD)**	
** 1. Operation**	140 (54.8)
** 2. Operation**	660.0 (313.1)
**Intraoperative transfusions**	1 Packed Red Blood Cells
**Overall hospital stay; days (SD)**	27.2 (7.7)
**90-day morbidity**	
** Clavien–Dindo > 3a (%)**	10
** 90-day mortality (%)**	0

**Table 3 cancers-16-01070-t003:** Comparison of the literature regarding patients who underwent complete robotic-associated liver partition and portal vein ligation (rALPPS). CR: case report, SC: single center, HCC: hepatocellular carcinoma, CRLM: colorectal liver metastasis, ICC: intrahepatic cholangiocarcinoma.

Author	Year	Country	Study Design	Diagnose	Cases (n)	Approach
Solomonov et al. [26]	2013	Israel	CR	HCC	1	rALPPS
Vincente et al. [27]	2016	Spain	CR	CRLM	1	rALPPS
Quijano et al. [28]	2017	Spain	CR	n.a.	1	rALPPS
Di Benedetto et al. [29]	2020	Italy	CR	HCC	1	rALPPS
Di Benedetto et al. [30]	2020	Italy	CR	ICC	1	rALPPS
Machado et al. [31]	2020	Brazil	CR	CRLM	1	rALPPS
Fernandes et al. [32]	2021	Brazil	SC	HCC	3	rALPPS
Hu et al. [33]	2021	China	CR	HCC	1	rALPPS
Arend et al.	2023	Germany	SC	CRLM/ICC	5	rALPPS

**Table 4 cancers-16-01070-t004:** Comparison of the literature regarding patients who underwent complete robotic-associated liver partition and portal vein ligation (rALPPS). OP: operation, FLR: future liver remnant, n.d.: no data. * Due to use of Aqua-Jet for parenchymal dissection, blood loss is invalid.

Author	Operation Time (min)/Mean		Intraoperative Blood Loss (mL)/Mean		Increase FLR (%)/Mean	Time OP 1.–2. (d)/Mean	Hospital Stay (d)/Mean
	OP 1	OP 2	OP 1	OP 2			
Solomonov et al. [26]	410	180	500	500	30	14	10
Vincente et al. [27]	n.a.	n.a.	n.a.	n.a.	43	13	n.a.
Quijano et al. [28]	540	180	n.a.	n.a.	n.a.	n.a.	64
Di Benedetto et al. [29]	495	280	480	200	91.6	10	13
Di Benedetto et al. [30]	470	380	150	300	112	14	14
Machado et al. [31]	293	245	420	270	46	21	8
Fernandes et al. [32]	446.7	276.7	n.d.	200	62.7	21	18
Hu et al. [33]	195	217	250	500	85	12	18
Arend et al.	341.2	440.6	140 *	660 *	39.9	17.8	27.2

## Data Availability

These are exclusively data in the system of the University Hospital Magdeburg. Public access to data is not possible and is not permitted for data protection reasons.

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
