# Peer review of "Robotic Complete ALPPS (rALPPS)—First German Experiences"

_cancers, 2024, doi:10.3390/cancers16051070_

Round 1

Reviewer 1 Report

Comments and Suggestions for Authors

I would like to congratulate the authors for sharing their first experience of robotic complete ALPPS (rALPPS) and for making it into a nice comprehensive publication. The manuscript contains enough technical surgical information and tips as well as data related to the perioperative course and outcomes. Perhaps, the paper could be improved by providing more precise data on patient positioning during the surgery, placement of trocars and robotic instruments in a schematic way. The literature analysis, discussion and the conclusions are sound and comprehensive.

-The main question addressed is the oncological safety of the robotic complete ALPPS (rALPPS) and possible improvement of morbidity and mortality.

-There is very little data about the outcomes of this novel surgical approach, thus all additional clinical data and technical tips are very needed.

-This manuscript reviews the experience and the results of the European high volume HPB/Cancer surgery center, thus allowing for much better benchmarking of the results in comparison to the previous clinical practice.

-Methodology is appropriate.

-The conclusions are in line with the presented results. Surely, all the readers will look forward to hear the results from a larger series of patients, however, this first experience is of critical importance.

-References are appropriate

Author Response

Many thanks for your Review,

Patients were placed in reversed Trendelenburg position. The robotic trocars were placed in the middle of the abdomen, approx. 15 cm below the costal arch, as described elsewhere [24,25]. The trocars are inserted in the right upper abdomen, to the right and left of the navel and in the left upper abdomen in a single line. One laparoscopic trocar for table assistance and pringle maneuver were placed between robotic trocar 2 and 3 and in the left lateral abdomen. 

Reviewer 2 Report

Comments and Suggestions for Authors

The manuscript by Arend and colleagues is a case series of fully robotic ALPPS with a literature review. This series represents a large cohort. 

The presentation is clear and the operative technique is clearly reported.

I noted that the review of previous cases is missing one case. (Masetti https://doi.org/10.1089/lap.2020.0589) the authors should add in table 3/4.

What is the authors' opinion on the difference between open and robotic times between the two phases. It seems that in the case of the minimally invasive approach the time is longer than in the open approach. The same results was reported in an Italian registry (10.1007/s00464-023-09937-4) 

Author Response

Many thanks for your Review,

First point:

In our article, we only describe cases with ALPPS without simultaneous colon resection. In our clinic, we perform colon resection approximately 3 weeks after complete ALPPS. This means that the data are not comparable with ALPPS and are therefore not listed in the table.

Second point:

The robotic time is longer compared to the open procedure. This primarily affects the second step. However, the surgical trauma is significantly less with the robotic procedure and leads to a faster convalescence

Reviewer 3 Report

Comments and Suggestions for Authors

In this paper by Arend J et al., the authors show the results obtained by exploiting the robotic Associating Liver Partition and Portal Vein Ligation (rALPPS) for resection of primary and secondary hepatic tumors showing that this technique can be carried out safely with reduced morbidity and mortality in selected patients.

This paper addresses the issue of ALPPS and the morbidity and mortality reported to be associated with it, showing that rALPPS allows to achieve a complete and safe oncological resection (R0).

The paper is well described and well documented with the information of patients and some pictures of the procedure, thus adding precious data for the evaluation and the improvement of the ALPPS, increasing the efficacy of the and reducing the morbidity and mortality associated with this technique.

To improve the paper, the authors should modify data shown in figure 2: it is not clear whether the data refer to one patient or all of them. In the case data are from all patients, is the figure showing average value or median value? Please specify this point and add error bars accordingly.

Additionally, what is the volume unit of measurement used (sl)?

Finally, even though the group of patients is limited in number, did the authors notice a different response associated with the pathology, i.e. colorectal liver metastases vs intrahepatic cholangiocarcinoma?

Comments on the Quality of English Language

Please check the manuscript for typos. e.g. Line 25: “rALPPS can be carried out safe with reduced morbidity and mortality in selected patients” correct with “rALPPS can be carried out safely with reduced morbidity and mortality in selected patients”

Author Response

Many thanks for your Review,

figure 2. shows the mean of all cases.

We have changed the labelling of figure 2.

Reviewer 4 Report

Comments and Suggestions for Authors

The authors described the experience performing rALPPS for colorectal liver metastasis and cholangiocarcinoma. These cases showed excellent liver regeneration and good outcomes. However, there are some limitations for publication.

1.       The tumor size with mean size of 45mm is relatively small in this study. The authors should describe the general indication for rALPPS and limitations up to data in the method or discussion. For large tumor, what size is indicated for rALPPS?

2.       This study utilizes LiMax value for liver function test. The authors should describe the indication regarding LiMax for rALPPS.

3.       The authors should refer to degree and frequency of post hepatectomy liver failure.

Author Response

Many thanks for your Review,

  1. The mean size of 45mm includes the largest tumour diameter for each patient. It does not take into account the summary of all tumour foci.
  2. We use the test for all liver resections (minor/major). There is no specific indication only for ALPPS.
  3. As this is a complex new procedure, comparison with other procedures in terms of postoperative liver failure is only possible to a limited extent. In the available cases in the literature research and in our own experience, there has been no postoperative liver failure to date.